# Emerging Trends in Advanced Translational Applications of Silver Nanoparticles: A Progressing Dawn of Nanotechnology

**DOI:** 10.3390/jfb14010047

**Published:** 2023-01-14

**Authors:** Shaheen Husain, Aditya Nandi, Faizan Zarreen Simnani, Utsa Saha, Aishee Ghosh, Adrija Sinha, Aarya Sahay, Shailesh Kumar Samal, Pritam Kumar Panda, Suresh K. Verma

**Affiliations:** 1Amity Institute of Nanotechnology, Amity University Uttar Pradesh (AUUP), Sector 125, Noida 201313, India; 2School of Biotechnology, KIIT University, Bhubaneswar 751024, India; 3Unit of Immunology and Chronic Disease, Institute of Environmental Medicine, Karolinska Institutet, 17177 Stockholm, Sweden; 4Condensed Matter Theory Group, Materials Theory Division, Department of Physics and Astronomy, Uppsala University, 75120 Uppsala, Sweden

**Keywords:** silver nanoparticles (AgNPs), synthesis, applications, toxicity

## Abstract

Nanoscience has emerged as a fascinating field of science, with its implementation in multiple applications in the form of nanotechnology. Nanotechnology has recently been more impactful in diverse sectors such as the pharmaceutical industry, agriculture sector, and food market. The peculiar properties which make nanoparticles as an asset are their large surface area and their size, which ranges between 1 and 100 nanometers (nm). Various technologies, such as chemical and biological processes, are being used to synthesize nanoparticles. The green chemistry route has become extremely popular due to its use in the synthesis of nanoparticles. Nanomaterials are versatile and impactful in different day to day applications, resulting in their increased utilization and distribution in human cells, tissues, and organs. Owing to the deployment of nanoparticles at a high demand, the need to produce nanoparticles has raised concerns regarding environmentally friendly processes. These processes are meant to produce nanomaterials with improved physiochemical properties that can have significant uses in the fields of medicine, physics, and biochemistry. Among a plethora of nanomaterials, silver nanoparticles have emerged as the most investigated and used nanoparticle. Silver nanoparticles (AgNPs) have become vital entities of study due to their distinctive properties which the scientific society aims to investigate the uses of. The current review addresses the modern expansion of AgNP synthesis, characterization, and mechanism, as well as global applications of AgNPs and their limitations.

## 1. Introduction

Nanotechnology is one of the most feasible innovations of our generation. Translating nanotechnology into useful implementations creates new innovations for simplifying people’s lives. Nanoscience and nanotechnology entail designing new materials, instruments, and processes based on atom regulation on a scale below 1 to 100 nm. Nanotechnology is a major contribution to research in many disciplines, including physics, materials science, chemistry, biology, computer science, and engineering. Increasingly, nanometer-size metal particles are observed to exhibit extremely different physical, chemical, and biological properties relative to their macro-sized counterparts. There have been many studies performed on nanoparticles in recent years [1,2,3,4]. Metallic nanoparticles contain properties that are of concern to scientists for different applications [5,6]. The synthesis and catalysis of nanoparticles requires an optimization of scale and form. The regulation of the scale and form of mass is essential. The specific regulation of particle size, form, and distribution is also accomplished by varying the synthesis processes and minimizing agents and stabilizers [4,7]. Metal nanoparticles can be prepared in three ways, through physical, chemical, or biological approaches. Under chemical reduction, metal ions in a solution are diluted chemically. Chemical methods can be subdivided into classical chemicals regarding the existence of the reduction agent; using the well-known chemical reducing substances (hydrazine, sodium borohydride, hydrogen, etc.); and radiation–chemical substances, in which the reduction mechanism is initiated by solved electrons produced by ionizing radiation [8]. However, chemical approaches may be classified into biological approaches, in which non-deleterious solvents and naturally occurring reductive agents are employed. Examples of such reductive agents include polysaccharides or plant extracts; biological micro-organisms such as bacteria and fungi as reductants; and those working in reverse-micellar systems, where the aggregation process takes place in the aqueous center of an inverted micel and growing particles [9]. The stabilization of nanoparticles is typically discussed as two basic types of stabilization: electrostatic and steric. Electrostatic stability is accomplished by coordinating anionic species, such as halides, carboxylates, or polyoxo-nions, with metal particles. Moreover, this process involves the creation of an electrical double layer, which causes coulombic repulsion between the nanoparticles. Steric stabilization is accomplished through the presence of bulky, usually organic materials; due to their bulk, these materials prevent the diffusion of nanoparticles together. Examples of steric stabilizers include polymers and large cations such as alkyl ammonium. The option of a stabilizer also helps to change the solubility of nanoparticles [10,11]. The usefulness of metal and metal ions, including silver and silver ions, has long been recognized. Silver and silver-based compounds (silver nitrate (AgNO_3_) and silver sulfadiazine (C_10_H_9_AgN_4_O_2_S)) have been used for a variety of uses in recent decades. However, their effectiveness in medicine has been considerably diminished by the production of multiple antibiotic compounds. Later, the discovery of variable-shape- silver nanoparticles (AgNPs) proved their significance as effective antimicrobial agents against various microorganisms. The antimicrobial and anti-inflammatory properties of AgNPs have been relevant in the medical sector [10,12] for regulating microbial infections. Moreover, the antimicrobial properties of AgNPs have been more effective as disinfectants and have also contributed to the treatment of endotracheal tubing, burns, intravenous catheters, and dental fillings [13,14]. AgNPs have shown considerable potential in the treatment of different forms of tumors, most notably cancerous cells. AgNPs could be used as contrast agents for photoacoustic imaging and computed tomography imaging for disease diagnosis or visualization. Thus, AgNPs are considered a recent trend in the production of therapeutic antimicrobials.

## 2. Traditional Techniques for the Synthesis of Metal Nanoparticles

The nanoparticle fabrication process involves a “bottom-up” approach or a “top-down” approach [15]. Particles of different sizes can be obtained during nanoparticle synthesis using different techniques (physical, chemical, and biological processes). The physical and chemical properties of nanoparticles are strongly dependent on the form and surface composition of the particles. Progress in the current era of nanotechnology and materials science is highly influenced by the novel “Green Synthesis” method for biocompatible nanomaterials. It has been acknowledged that natural substances such as cellulose and plant biomolecules can be used as sources for the creation of nanomaterials. The potential uses of nanomaterials developed from green synthesis in biological and environmental applications have been studied [16] (Figure 1).

### 2.1. Physical Method of Nanoparticle Synthesis

Physical methods of nanoparticle synthesis comprise UV irradiation, sonochemistry, laser ablation, and more. During a physical synthesis process, the metal atoms evaporate, followed by a condensation process on different supports, whereby the metal atoms are aggregated into tiny clusters of metallic nanoparticles. Using this method, we can generate nanoparticles of high quality and with specified shapes. A traditional laser requires extraordinarily complex equipment, materials, and heavy power consumption, contributing to high manufacturing costs [17]. Physical methods of synthesizing nanoparticles have the advantage of producing bulk amounts of nanoparticles; however, the limitation of controlling shape and size exists.

### 2.2. Chemical Method of Nanoparticle Synthesis

Another method for synthesizing nanoparticles is to use metal ions rather than chemicals. Depending on the reaction conditions, metal ions may either form small clusters or precipitate into larger clusters. Hydrazine, sodium borohydride, and hydrogen are used as reduction agents [18]. Often-used reduction agents are micelles of synthetic or natural polymers such as cellulose, natural rubber, chitosan, and co-polymers. These compounds include other organic solvents, such as ethane, dimethyl, formaldehyde, toluene, and chloroform. Hazardous materials are of a toxic origin and cannot be recycled. To mitigate the problems of manufacturing metals in this manner, scientists and researchers are studying nanoparticle-processing methods. Chemical methods of synthesis have been to be proved some of the most-used tools for the production of nanoparticles; however, the hazardousness of chemicals in ecotoxicological and biomedical aspects is a limitation.

### 2.3. Biological Method of Nanoparticle Synthesis

The biogenic synthesis of metallic nanoparticles has recently achieved international fame. The synthesis of nanoparticles by microorganisms and plants is accomplished in biogenic synthesis processes. These processes can generate nanoparticles of a better-defined size and morphology than some other physicochemical synthesis methods. New technology proposes new means of synthesis for nanoparticles, such as sluggish kinetics, a lowered financial investment on enzymes, and plants that act as biological templates [19]. Nanoparticles can be engineered using biological methods in the absence of toxic, expensive, and harsh chemical substances [1,20]. In comparison to other techniques, green synthesis has a very bright future in the open market for obtaining nanostructured materials. A bottom-up method called the “green synthesis” of nanoparticles (NPs) produces NPs through the oxidation/reduction of metallic ions by organic molecules derived from biological sources. We understand that when the term “green synthesis” is used, it implicitly refers to a cheap and straightforward procedure without the use of expensive or complicated equipment. These characteristics have made green synthesis stand out in the field of developing and expanding techniques for metallic nanostructures.

**Figure 1 jfb-14-00047-f001:**
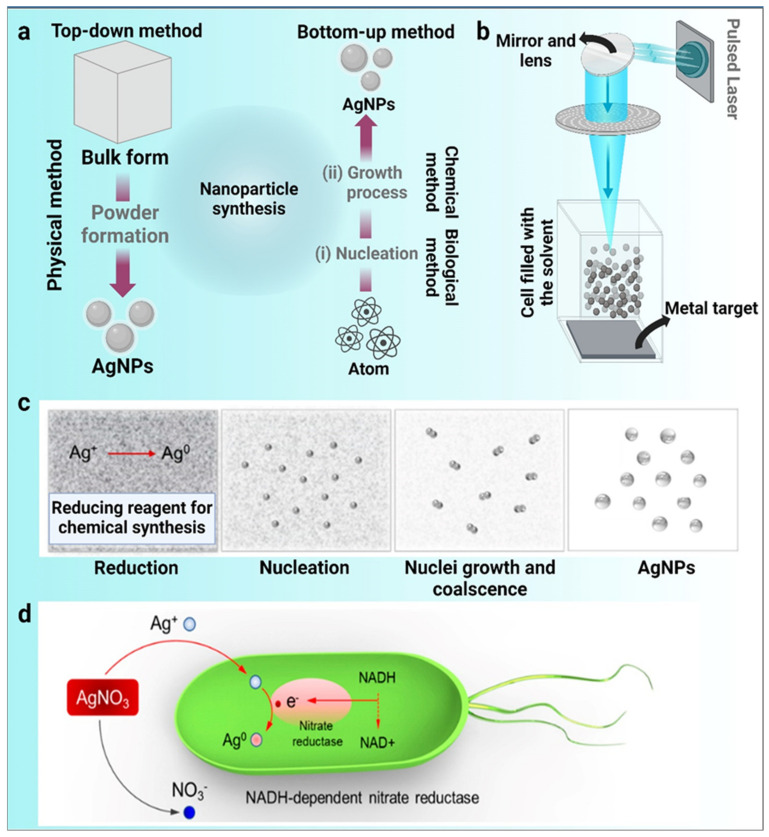
Diverse synthesis routes of silver nanoparticles (AgNPs). (**a**) Top-down and bottom-up methods, (**b**) physical synthesis method, (**c**) chemical synthesis method, and (**d**) plausible synthesis mechanisms of green chemistry. The bio-reduction is initiated by the electron transfer through nicotinamide adenine dinucleotide (NADH)-dependent reductase as an electron carrier, forming NAD^+^. The resulting electrons are obtained by Ag^+^ ions, which are reduced to elemental AgNPs (Images adapted and modified from [21]).

## 3. Characterization of Nanoparticles

Nanoparticles are well known for their dispersity, size, surface area, and shape. Their uses are numerous; the uniformness of the nanoparticles’ properties is crucial. The affirmation of the engineered materials is performed by following several material-characterization processes. The characterization of nanoparticles is performed using some universal techniques, which involve UV–visible spectrophotometry, dynamic light scattering (DLS), scanning electron microscopy (SEM), transmission electron microscopy (TEM), energy dispersive spectroscopy (EDS), Fourier transform infrared spectroscopy (FTIR), powder X-ray diffraction (XRD) [15], and UV–visible spectroscopy [22]. Characterization of the AgNPs with spectrophotometric absorption measurements is performed to measure the size distribution and the surface charge of the particles suspended in a liquid. An additional method of characterization is electron microscopy [23], with the SEM and TEM techniques belong to such a method. Commonly, both SEM and TEM come with the unit of an EDS system, which assists in detecting the elemental composition of materials [20]. When compared to scanning electron microscopy, TEM scanning electron microscopy provides a much better resolution of the images. The field emission scanning electron microscope (FESEM) is an outcome of the revolution in the SEM technique, in which a greater energy range is applied and images with enhanced resolution can be obtained [24]. On the other hand, the FTIR technique is adapted for the characterization of surface chemistry, as the organic functional groups and other surface chemical residues attach to the surface of nanoparticles [25,26]. Similarly, the XRD technique is used to investigate the phases of nanoparticles [27].

## 4. Applications of AgNPs

The atomic mass of silver (Ag) is 107.87 amu and its atomic number is 47. Silver has been used as a therapeutic agent for hundreds of years. In classical Indian Ayurvedic science, silver is used to encourage good health. It is commonly accepted that AgNPs are more powerful than pure silver. The surface area of Ag is increased significantly when it is transformed into AgNPs, which increases its biological and optical properties. Some of the prominent applications of AgNPs are discussed below.

### 4.1. Medical Applications of AgNPs

Researchers are progressively showing a keen interest in the biomedical applications of AgNPs. These applications lay emphasis on the potential use AgNPs to cure many diseases and can therefore be a boon to medical industries (Table 1). Scientists are focusing on producing conductive AgNPs that have an ideal size and morphology. In this study, the emphasis is on the relevance and applications of AgNPs (Figure 2).

#### 4.1.1. AgNPs as Anti-Bacterial Agents

The ecosystem posed a serious threat to human health when bacteria became immune to antibiotics [28]. The production of novel antibacterial agents is important for this cause. AgNPs have been widely researched for their antibacterial properties against bacteria such as *P. aeruginosa*, *P. vulgaris*, *E. coli*, *S. aureus*, etc. [29]. A recent investigation by Verma et al. reported the inverse cytotoxicity and enhanced antibacterial activity of the green-synthesized AgNPs [30]. The effect of AgNPs on microorganisms comes from several theories: the inhibition of the enzymes that cells need for survival, the activation of enzymes that cells need for survival, the diffusion of silver ions into cells, and an increased cell permeability. There is a risk that AgNPs might be lethal to bacterial cells; another discourse surrounding the behavior of AgNPs is that they damage the bacterial cell membrane. As AgNPs can be extracted from nanoparticle sources of silver, these particles can impact enzymes and proteins, in addition to regulating their activity. Bacterial cells contain reactive oxygen species (ROS) which suppress enzyme functions in the process of cell division. The interaction of the phosphorus and sulfur of bacterial DNA with AgNPs causes an end product of microbial death. The dephosphorylation of tyrosine utilizing NPs causes the inhibition of signal transduction and thus blocks the growth of microbes such as *Escherichia coli* and *Staphylococcus aureus*, etc. [31,32].

#### 4.1.2. Implications of AgNPs in Catheters

Catheters are instruments that are implanted in the body to cure medical conditions or perform surgery. By adapting and changing their configuration, it is possible to build catheters for various medical applications. Urinary catheters are used for several purposes, such as diagnostic, clinical, tracking, and ease of patient treatment. To avoid urinary blockage, a short-term catheter can be used intermittently, or it can be left in place for a prolonged period. Long-term use of urinary catheters induces UTI. Catheter-induced urinary tract infections (CAUTI) in hospitals are caused by both Gram-positive and Gram-negative bacteria. Nanoparticles, being anti-microbial agents due to their higher surface-to-volume ratio and novel physical and chemical properties, have been used to reduce catheter-borne infections. The research indicates a reduced risk of urinary tract infections when using silver-nanoparticle-coated catheters [33,34]. Plastic catheter tubes consisting of AgNPs are extremely efficient antibacterial catheters. In vitro results demonstrated the efficacy of the newly synthesized compounds against biofilm growth as well as a longer period of effect. AgNPs were also tested to see if silver nanoparticle catheters were equally successful as regular catheters. The study found that just five out of the sixty-eight controls were positive for catheter-associated ventriculomeningitis (CAVM/CAV). In the community of silver-nanoparticle-coated catheters, no cases of central venous catheterization (CVC) occurred, and all central blood culture findings were negative. It has been recommended that AgNPs are potentially useful in the prevention of cavities and are encouraged for humans at large with no records of toxicity. The hope is that this example will help repair the image of AgNP-impregnated venous catheters for clinical use. Catheters are used in neurosurgery to alleviate discomfort in the brain by eliminating extra cerebrospinal fluid. Neurosurgical catheters may be indefinitely inserted and used as shunts. Although studies have reported AgNPs to be a successful candidate in catheter technology, more investigations are required for precise verification.

#### 4.1.3. AgNPs in the Detection of Alzheimer’s Disease

Alzheimer’s disease (AD) is an autoimmune disease that can be recognized with AgNPs. There has been a massive influence of LSPR (localized surface plasmon resonance) technology in the pitch of nanoscience-based diagnosis of AD [35]. LSPR is a nanosensor technology that detects molecular biomarkers of AD and is focused on the characteristics of triangular AgNPs. This type of material operates upon magnetic refraction and magnetostriction. This change alters the AgNPs’ λ_max_, which could be detected via spectroscopy. The LSPR nanosensor is also susceptible to multiple concentrations of a target biomolecule. These LSPR nanosensors support the research on Alzheimer’s disease and study the pharmacological interactions between drugs and their targets [36].

Scientists found that neural cells can uptake 3–5 nm of AgNPs. They have investigated the potential effects of AgNPs on the gene expression of inflammation and neurodegenerative disorder in murine brain ALT astrocytes, microglial BV-2 cells, and neuron N2a cells. This study, conducted by Huang et al., revealed that post-exposure of AgNPs to mouse brain cells led to an upsurge in IL-1β secretion in neural cells and induced the gene expression of C-X-C motif chemokine 13 (CXCL13). It was also found that the gene expression of amyloid precursor protein (APP) was induced, and the presence of neprilysin (NEP) and low-density lipoprotein receptor (LDLR) was reduced in neural cells. These results suggested that AgNPs could alter gene and protein expressions of Aβ deposition to potentially induce AD progress in neural cells [37]. Future investigations may pave the way for the broad-aspect application of AgNPs in curing Alzheimer’s disease.

#### 4.1.4. Dental Applications of AgNPs

Nanoparticle-based therapies have been documented to stimulate the re-mineralization process and monitor the formation of biofilm to reduce or remove the effect of dental caries [38]. Biofilms on dental implants can lead to infections in the host’s oral mucosa which, in turn, can lead to implant failure [39]. Silver has a fascinating background in oral care. Due to its application as an amalgam used for tooth preservation, it has attracted popularity all over the world [40]. AgNPs are used as dental restoratives, braces, endodontics, nanomaterial-related restoratives, as a regenerative, and as a multifunctional medicine [41,42]. Their use is combined with normal dentures for additional bacterial-destroying skills. In addition, AgNPs were also found to inhibit dental staining due to their decreased size and increased surface area; this was observed when silver was used as nanosilver diamine fluoride [43,44]. Therefore, antibacterial resins used in orthodontics and restorative dentistry have been incorporated with AgNPs in acrylic-resin, denture-based products to enhance their physicomechanical properties and antimicrobial impact [45]. Some studies also designated the potential for AgNPs to be used as biostatic or biocide coatings for traditional, titanium-based dental implants [46]. AgNPs may also be theoretically effective in dental applications if they compete against cytotoxicity.

#### 4.1.5. Orthopedic Applications of AgNPs

The treatment of osteomyelitis is costly, and the rise in antibiotic-resistant bacteria leads to prolonged hospital stays. The prevalence of osteomyelitis has risen with technical advancements because of the expanded use of orthopedic implants in operations [47]. The reported deep-tissue infection rates are 0.5–17% after intramedullary nailing and joint substitution [48]. Several treatment approaches for osteomyelitis have been identified, each of which has various benefits and side effects. Orthopedic implants coated with AgNPs have been developed, and the effects of AgNPs on bone cells have been explored as AgNP-coated orthopedic implants are a promising method for preventing infection. AgNPs are toxic to osteoblasts [49,50,51]. Specialized, bone-forming cells and osteoclasts resorb the bone [49]. Moreover, controversial findings are available on the multipotent precursors of osteoblast mesenchymal stem cells (MSCs). The AgNPs effect the dosage and time-dependent viability of MSCs, potentially via the induction of DNA damage [52,53]. Since postoperative infections of orthopedic implants can lead to fixation failure, and because resistant bacteria are rapidly emerging in these devices, AgNPs might represent a good candidate for coating orthopedic implants and thereby circumventing bacterial colonization. The use of AgNPs, however, is addressed in vivo and various cell types because of their toxicity [54,55,56,57]. AgNP-constructed bone cement has not been cytotoxic in mouse fibroblasts or human osteoblasts, indicating their biocompatibility. However, the fundamental processes are not understood in many instances. It is reported that the co-substitution of titanium dioxide (TiO_2_)/Ag-containing hydroxyapatite exhibited significant synergistic, long-term bactericidal properties in vitro [58,59].

#### 4.1.6. Role of AgNPs in Wound Healing

Current therapeutic use of AgNPs includes their use in wound dressings for the treatment of multiple wounds such as pemphigus [60], toxic epidermal necrolysis [61], Steven–Johnson syndrome [62], burns [63], and chronic ulcers [64]. Two layers of polyethylene maze that form a sandwich around a polyester gauze sheet are typical dressings [65] in which AgNPs are used. The polyethylene layer is coated in AgNPs (10–15 nm) in the 900 nm-thick layer. The superior wound-healing properties of AgNP dressings have been tested in randomized clinical trials (RCTs) over current silver sulfadiazine and burning gauze dressings. Dressings with AgNPs significantly decreased the wound-cure time and improved bacterial removal from infected wounds. AgNPs demonstrate in vitro skin permeation,; this increases in the case of damaged skin [66]. With modern nanotechnology methods, biopolymers (e.g., chitosan or collagen) can create stronger platforms for a successful wound-healing method. Examples of some of the approved Food and Drug Administration (FDA) (US) biocomposites modified for wound dressing applications using ionic silver are the Acticoat^TM^, Bactigras^TM^, Aquacel^TM^, Poly Mem Silver^TM^, or Tegaderm^TM^. In addition, new, naturally derived biomaterials (cotton fibers, bacterial cellulose, sodium chitosan, etc.) were also introduced into AgNPs to improve the management of wound healing.

#### 4.1.7. AgNPs as an Anti-Inflammatory Substance

AgNPs has been studied as a potent agent for anti-inflammatory actions. Studies have shown the action of AgNPs to inhibit 1,2-Dinitrochlorobenzene-mediated touch dermatitis in swine [65]. Immune–histochemical studies have shown the higher concentrations of TGF-B and TNF-a, pro-inflammatory cytokines in human cells, upon exposure to AgNPs rather than silver nitrate. The anti-inflammatory action of AgNPs, such as cytokine-reduction activity, has been theorized to be mediated by a decrease in lymphocyte- and mast-cell-infiltration for inducing apoptosis in inflammatory cells [65,67]. The role of proteinases of matrix metal (MMPs) has been shown to induce chronic ulcers instead of toxoids, indicating the nature of MMPs to improve the chronic ulcers’ non-healing nature. Nano-crystalline dressings using AgNPs have shown a decrease in MMP-9 in a pig model, leading to an enhanced wound cure [68]. Moreover, the use of AgNPs in wound dressings facilitated the healing of stalled, recurrent leg ulcers in a clinical human trial (n = 15); this has been attributed to the anti-inflammatory nature of AgNPs and their antibacterial actions [64].

#### 4.1.8. Anti-Fungal Properties of AgNPs

Multi-functional AgNPs, when used in therapy against spores, may be an important solution against potentially harmful fungal growths. A fungal cell membrane was treated with metallic nanoparticles, which caused alterations in the fungal cell membrane structure [69]. The fungicidal mechanism of biosynthesized metallic nanoparticles has more potential than commercial antibiotics such as fluconazole and amphotericin. The plant-derived AgNPs clearly showed membrane damage to *Candida* sp. and in fungal intercellular components and, finally, cell function was destroyed [70]. The antimicrobial properties of silver have been recognized and used as a standard treatment for bacterial skin infections caused by *Staphylococcus aureus* and *Pseudomonas aeruginosa* [71]. On the other hand, AgNPs have shown a broad-spectrum antimicrobial activity, including against fungal agents of opportunistic infections [72] such as *Candida albicans*, *C. tropicalis*, *C. parapsilosis*, *C. glabrata* [73], *Trichophyton rubrum* [74], *Trichosporon asahii* [75], *Aspergillus niger*, *Rhizoctonia solani*, *Curvularia lunata*, *Colletotrichum* sp., and *Fusarium* sp. [76].

#### 4.1.9. Anti-Plasmodial Properties of AgNPs

Insects such as mosquitoes are the primary vector for the spread of malaria, dengue fever, yellow fever, filariasis, schistosomiasis, Japanese encephalitis, Lyme disease, and West Nile, etc., causing millions of deaths per year [77]. The AgNPs have been recommended as a potent larvicide against the *Aede aegypti* larvae [78,79]. Compared to the other nanoparticles, such as gold NP, AgNPs have been found to be more successful against the larval phases of the mosquito. A study demosntrated the significant action of silver nanoparticles on the larva of *Anopheles subpictus*, *Culex quinque fasciatus*, and *Ripicephalus microplus* in coordination with the aqueous leaf extract of *Mimosa pudica* [80]. AgNPs synthesized using the aquatic leaf extract of *Tinospora cordifolia* demonstrated a successful impact on *Anopheles subpictus* and *culex quinquefasciatus*, *pediculosis* [81]. The large-scale effect of aqueous crude leaf extracts, silver nitrate, and the synthesized AgNPs of *Mimosa pudica* have also been shown to exhibit the highest mortality rate with the *Anopheles larvae* and *Culex quinquefasciatus* [80]. AgNPs were also recorded to have significant action against late-third larval age group of *Vulgaris* and *C. quinquefasciatus* [82,83].

#### 4.1.10. Anti-Cancer Possessions of AgNPs

The modern uses of metallic nanoparticles to detect and cure different forms of cancer in the medical sector have been shown. Most recent experiments have shown that nanoparticles extracted from plants could regulate the growth of tumor cells. New and revolutionized organic nanoparticles are used to treat malignant deposits without mixing with normal cells. A recent finding by Suman et al. recorded a major cytotoxic effect for HeLa cell lines by biosynthesized AgNPs relative to other synthetic chemical drugs [84]. The cytotoxic impact was strengthened due to the presence of secondary metabolites and other non-metal compositions [85,86]. The AgNPs synthesized from plant entities were found to pedal the cell cycle and bloodstream enzymes. In addition, the plant-synthesized nanoparticles regulated the production of free radicals from the cell. Free radicals typically cause the proliferation of cells and disrupt normal cell. Studies have shown the good anticancer activity of *Salacia chinensis* extract-channelized AgNPs (40–80 nm) for liver (Heep-G2), cervical (HeLa), lung (L-132), pancreas (MIA-Pa-Ca-2), breast (MDA-MB-231), oral (KB), and prostate (PC-3) cell lines with IC50 values of 6.31, 6.55, 4.002, 5.228, 8.452, 14.37, and 7.46 μg/mL, respectively. Similarly, Husain et al. showed the anticancer activity of cyanobacterial, synthesized AgNPs against MCF-7 and HepG2 cell lines [3]. AgNPs also play important roles as photoactive agents, drug delivery carriers, or radiosensitizers in cancer treatments.

#### 4.1.11. Anti-Viral Properties of AgNPs

Plant-mediated nanoparticles are alternative drugs for treating and controlling the growth of viral pathogens. It was found that biogenic AgNPs act as potent, broad-spectrum MCF-7 antiviral agents. Suriyakala et al. found that the bio-AgNPs demonstrated persuasive anti-HIV action at an early stage of the reverse-transcription mechanism [87]. Studies have also shown the capacity of PVP-coated AgNPs to disrupt the interactions between the HIV glycoproteins (gp120) [88]. Similarly, Du et al. created electrostatic, self-assembled, glutathione-capped, silver sulphide nanoparticle (GO-AgNPs) nanocomposites for a potent antiviral agent. Porcine reproductive and respiratory syndrome virus (PRRSV) and porcine epidemic diarrhea virus (PEDV) were the targets of the research to evaluate the antiviral activity of the GO-AgNPs [89]. According to their findings, the GO-AgNPs nanocomposites prevented PRRSV and PEDV infection by lowering the expression of the viruses’ nucleocapsid proteins, which in turn prevented virus replication. They also found that the GO-AgNP nanocomposites stopped PRRSV and PEDV from entering host cells. The metallic nanoparticles are strong antiviral agents and inhibit viral entry into the host system. An investigation reported that the biosynthesized metallic nanoparticles have multiple binding sites to bind with the gp120 of the viral membrane to control the function of the virus [90]. In light of this, it can be said that the biogenic or hybrid (physical and biochemical) synthesis of AgNPs and nanocomposites could be very advantageous as a broad-spectrum antiviral agent for the suppression of numerous pathogenic viruses.

#### 4.1.12. Anti-Diabetic Properties of AgNPs

The biosynthesized AgNPs can also be regarded as alternative agents for curing diabetes mellitus. Swarnalatha et al. explored that the *Sphaeranthus amaranthoides* biosynthesized AgNPs inhibited α-amylase and sugar in diabetes-induced animal models [91]. Likewise, Pickup et al. studied that nanoparticles are potent therapeutic agents for controlling diabetes [92]. AgNPs have been reported to effectively inhibit carbohydrate-digesting enzymes such -amylase and -glucosidase, with IC_50_ values of 54.56 and 37.86 g mL^−1^, respectively, demonstrating their anti-diabetic properties [93]. Findings are still limited; henceforth, the performance of more studies is required to assure the effectiveness of AgNPs as anti-diabetic agents.

**Table 1 jfb-14-00047-t001:** Biomedical applications of AgNPs.

S.No.	Source of Synthesis	Size Range of AgNPs (nm)	Biomedical Applications	References
1.	Cyanobacteria	7 nm	Liver cancer, breast cancer	[3]
2.	*Cucumis melo* L. (muskmelon)	25 nm	Anticancer, antibacterial	[94]
3.	*Benincasa hispida*	26 ± 2 nm	Antibacterial, human cervical cancer	[95]
4.	*Lactobacillus brevis* MSR104	45 nm	Antibacterial, anticancer, antioxidant	[96]
5.	*Spirulina platensis*	29 nm	Biofilm formation in urinary catheters	[97]
6.	N,N,N′,N′-tetramethylethylenediamine	100 nm	Catheter-related infections	[98]
7.	Flavonoids, phenolic compounds, and glucose extract	9.1 ± 0.4 nm	Catheters	[99]
8.	*Commercially purchased*	20–40 nm	Bone repair, antibacterial, wound healing	[100]
9.	*Rosa indica wichuriana* hybrid leaf extract	15–20 nm	Antifungal, anti-bacterial	[101]
10.	*Immobilization*	16 nm	Antibacterial	[102]
11.	Chemically synthesized from NaBH_4_	5–500 nm	Biosensor for the detection of human chorionic gonadotropin	[103]
12.	Tannic acid	13,33,46 nm	Antiviral	[104]
13.	*Artemisia annua*	50 nm	Anti-malarial	[105]
14.	*Commercially purchased*	20–30 nm	Anti-Giardia	[106]
15.	*Indigofera oblongifolia*	10–30 nm	Anti-plasmodium	[107]
16.	*Commercially purchased*	7 ± 4 nm	Coating on appliances	[108]
17.	*Eucalyptus camaldulensis*	4–30 nm	Coated on braided silk, surgical	[109]
18.	*N. khasiana*	10–15 nm	Treatment of Alzheimer’s disease	[110]
19.	*Egg white*	9.5 ±2 nm to 30.2 ± 2 nm	Wound healing	[111]
20.	Sodium borohydride	15 nm	Orthopedics	[112]
21.	Acacia nilotica or natural gum	10–78 nm	Different cancer cell lines	[113]
22.	Chitosan	17–50 nm	Cancer cell line HepG2	[114]

### 4.2. AgNPs as an Odor-Controlling Agent in the Textile Industry

Researchers have focused on the process of AgNP synthesis with a regulated size and shape in order to find the application of AgNPs in textiles; not just on the surface of textile fibers, but also on their composition [5,115,116,117]. Typically, ex situ and externally implanted AgNPs exhibit low-energy adhesion to the fibers. The in situ amalgamation of AgNPs on textile fibers is also established to address the recurring issue of conformity in textiles [5,115,117]. Many other studies have been performed based on the surface analysis of textiles impregnated with AgNPs and other nanoparticles to evaluate their flexibility against different washings in an alkaline bath and their antibacterial effectiveness [118,119,120,121,122]. In recent study, basic research was carried out on the on-site growth of AgNPs on five types of textile surfaces including cotton, sheep wool, polyamide, and polyester that had various surface properties, such as hydrofoamics, rawness, and porosity. Two concurrent reaction phases were defined: first, the fibers were immersed in a silver ion solution (AgNO_3_ g^−1^·L^−1^), followed by their immersion in a chemical reduction solution. Scientists concluded that the strong chemical association of the AgNPs in wool-fiber ions—namely, a strong connection with pre-existing functional groups carboxyl (–COOH), amino (–NH_2_), and hydroxylate (–OH)—can be attributed to the strong physical–chemical adhesion of the AgNP in situ with the fibers [122,123]. The insertion of AgNPs into cotton fibers, particularly using cotton fabrics and various physicochemical methods, has been widely investigated [115,117] (Figure 2). In this respect, the poor adhesion of AgNPs to cotton fibers remains the key restriction since a large amount of AgNPs are released during the first wash cycles [122]. In longer durations, the concentration of AgNPs may play an essential role. Woolen fibers, using the process of in situ chemical reduction, have an ideal surface property to attract AgNPs with a high adherence efficiency. The antimicrobial properties of AgNPs are primarily mediated by the release of Ag+ ions from AgNPs, which enter bacterial cells and inhibit cell metabolism, as is widely defined by different studies [124,125].

**Figure 2 jfb-14-00047-f002:**
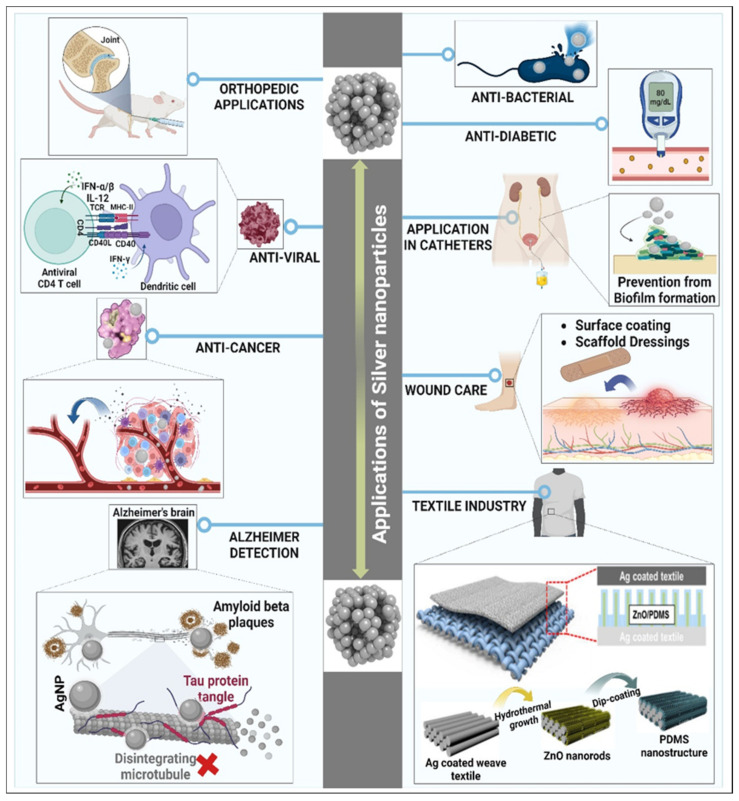
Biomedical and textile industry applications of AgNPs. (Image adapted and modified from [126]).

### 4.3. AgNPs in Cosmetics

AgNPs are also commonly used in the food and personal care sectors. Commercial products such as sunscreen, hair shampoo, soap, and detergents are known to contain AgNPs as a constituent. AgNPs are also reported to prevent skin diseases such as atopic dermatitis [127]. Although the effective action of AgNPs against skin infection is well known, there is a limitation of information about the exact mechanism. It is proposed that the bacterial cell wall may be damaged by the action of AgNPs. There are no side effects on human wellbeing at low and reasonable silver concentrations. AgNPs can be used as a preservative in cosmetics, and for anti-acne preparations due to their antibacterial effects. AgNPs are also formulated as preservatives in tubes of toothpaste and shampoos. Kim et al. have shown that AgNPs inhibit dermatophyte formation, making them a possible anti-infective agent [128,129]. Due to its antibacterial, anti-flu, and cancer-inhibitory effects, nanosilver (a form of AgNP) is also used in dietary supplements. Research shows that there may be variation in the antibacterial and antifungal effects of AgNPs according to their shapes, sizes, and forms. Although the antimicrobial action of AgNPs is due to the release of silver ions, the literature indicates that they may also have external effects that cannot be explained purely by the release of silver ions into the solution. There is concern over the possible penetration of nanomaterials into the skin as cosmetics are applied. Formulators for skin and hair treatments take due care that the products are supplied to the right locations. In fact, the skin is semi-permeable; ittherefore would not allow even nanomaterials to move easily through it. The study specifically concludes that nanoparticles do not pass into human skin from current cosmetic use, except in situations where the skin is compromised [130]. In a study by Kokura et al., it was shown that Ag nanoparticles are not capable of human skin penetration [131]. However, AgNPs on the skin surface will enter the skin when the barrier function of human skin is compromised. The probability of AgNP skin penetration was found to be from 0.2% to 2% (0.002–0.02 ppm); AgNPs did not demonstrate any toxicity at these stages [131]. Intact or partly impaired skin-touching nanoparticles (20 to 200 nm) do not reach the skin layer to penetrate the lower strata, making them healthy cosmeceuticals.

The deeper layer of the stratum corneum could be entered by nanoparticles with a diameter less than 10 nm, while NPs with a diameter greater than 40 nm could only reach 5–8 μm through the stratum corneum [132]. Chromium, platinum, TiO_2_, and ZnO nanoparticles do not penetrate further than the stratum corneum [133]. Acute dermal toxicity tests in Sprague Dawley rats on the formulation of silver nanoparticle gel (S-gel) demonstrated full protection for topical use. These findings clearly show that AgNPs in the form of a topical antimicrobial formulations may provide a safer alternative to conventional antimicrobial agents. Nanoscale calcium phosphate (apatite), used in certain special tooth creams for the neck of sensitive teeth, creates a thin coating similar to normal tooth enamel, thereby reducing pain sensitivity. Tiny fragments of nanometer-thin pigment can be contained in makeup, and gold and AgNPs are used in some day and night creams to give the skin a fresher look [134]. The luxury makeup line GNS Nanogistnanover^TM^ includes nanosilver, and the range of GNS Nanogistnanover^TM^ Q10 also includesnanosilver [134]. Nanosilver is found in soaps, toothpaste, wet wipes, lip cosmetics, deodorants, and foams for the face and body. It was reported that nanosilver-containing cleansing soap had bactericidal and fungicidal properties and was found to help treat acne and sun-damaged skin [135]. Essential metrics include a high effectiveness to deter the spread of infectious diseases within a limited exposure period. In hand wash, nanosilver at a concentration of 15 mg per liter was found to fulfill all criteria, and it helps destroy the bacteria very effectively [136]. Researchers have also shown that AgNPs can be used to kill yeasts such as *Candida glabrata* and *Candida albicans*, which allows nanosilver to kill mouth infections and be inserted into dentifrices [134]. Therefore, AgNPs are used in veterinary, medicinal, and biological materials. For burn victims, nanosilver skin cream, which contains thirty times less silver than silver sulfadiazine, is a safer option for the treatment of infections [137].

## 5. Role of AgNPs in Agriculture

### 5.1. AgNPs in Crop Production

Seed-germination inhibition is a common toxic effect of metal-based nanoparticles. El-Temsah and Joner studied the toxic effect of zero-valent iron NPs and AgNPs differing in average particle size from 1 to 20 nm on ryegrass, barley, and flax seed germination when exposed to 0–5000 mg L^−1^ of zero-valent iron NPs or 0–100 mg L^−1^ of Ag [138]. The study showed inhibitory effects in aqueous suspensions at 250 mg L^−1^ for zero-valent iron nanoparticles. A complete inhibition of germination was observed at 1000–2000 mg L^−1^ for zero-valent iron nanoparticles. AgNPs inhibited seed germination at lower concentrations but showed no clear size-dependent effects and never completely impeded germination. However, a positive effect of AgNPs on seed germination has also been observed. Studies suggest that metal AgNPs increase plant growth and development. Applications of AgNPs resulted in an increase in plant height [139], leaf number [140], weight of roots [141], plant biomass [139,140], seed germination [142], seed yield [139,140], quality of fruit [143], stem length, the diameter of canopy area [144], root length [142,145,146], growth, and the development of explants under in vitro culture conditions [147]. The addition of an appropriate nanosilver concentration increased chlorophyll content [148], carotenoids [149], flavonoids [147], photosynthetic quantum efficiency, enzymatic activity, the content of parahydroxy benzoic acid [150], α-terpinyl acetate, and antioxidative enzyme activity [144].

### 5.2. AgNPs as Herbicides

In agriculture, pesticides and herbicides are typically used to achieve better crop yields and production. Today, however, the harmful environmental implications of synthetic pesticides and herbicides are under debate. The key drawbacks of pesticides are pathogen- and pest-resistant growth, nitrogen-fixation reduction, soil biodiversity loss, pesticide bioaccumulation, pollinator decline, and habitat degradation for birds. Therefore, the substitution of NPs for herbicides addresses this issue to the utmost degree by reducing the number of herbicides needed for weed eradication. Herbicides are released into the soil according to the soil state with the active ingredient and smart distribution system. AgNPs have a pesticide action against pathogenic fungi and have been reported to have inhibitory effects on the conidial germination of the *Raffaelea* genus, causing oak tree mortality [151]. New methods of enhancing the stability of biological agents can be provided with the use of nanoformulations. The green synthesis of AgNPs of *Tinospora cordifolia* showed a maximum tidal effect against the head louse *Pediculus humanus* and the fourth-instar larvae of *Anopheles subpictus* and *Culex quinquefasciatus* [81]. Researchers have recently concluded that nano agrochemicals are typically nano-reforms to current pesticides and fungicides. Nano formulations are likely to increase the solubility of poorly soluble active components, release the active components in a targeted way, and avoid premature degradation [152]. Pesticide distribution, low chemical dosage, and high positive outcomes were substantially regulated by nano-pesticides. AgNPs at 100 mg kg^−1^ reduced mycelium growth and conidial germination against powdery mildew on cucurbits and pumpkins [153].

### 5.3. AgNPs as Pesticides

The adult hematophagous flies *Hippobosca maculate* and cattle ticks *Rhipicephalus microplus* were killed by AgNPs between 5–25 mg L^−1^, respectively [154]. AgNPs were also reported to have dose-dependent action against larvae in another study of *Rhipicephalus microplus* between 1.25–20 mg L^−1^ [81]. Other AgNPs exhibited dose-dependent activity against *Hippobosca maculate* and *Haema physalisbispinosa* between 2–10 mg L^−1^ [155] (Figure 3).

### 5.4. Improvement of Soil Quality through AgNPs

Soil damage entails unnecessary soil tilling. This leads to erosion and irrigation without drainage being required, which can be considered a route of salinization. This process serves the needs of human beings for bread, animal feed, and fiber. Long-term studies are needed to demonstrate the effects of various activities on sustainability-critical soil properties and to provide valuable evidence for this goal. In the United States, the development of nano-chemicals have emerged as promising agents for plant growth and pest control. Nanomaterials function as fertilizers with characteristics such as seed enhancement and less eco-toxicity. Plants may provide an essential route into the food chain for their bioaccumulation (Figure 3).

Recent advances in agriculture include the use of nanoparticles for the more efficient and effective use of plant chemicals. Several workers have documented the impact of various nanoparticles on plant growth and cytotoxicity. This included the impact of magnetite (Fe_3_O_4_) nanoparticles on plant growth; the impact of alumina, zinc, and zinc oxide on the seed germination and root growth of five higher plant species—radish, rape, lettuce, corn, and cucumber—as well as the impact of AgNPs on seedling growth in wheat [156], the impact of sulfur nanoparticles on tomatos [157]; the presence of zinc oxide in mungbeans; and the presence of nanoparticles of AlO (Aluminium(II) oxide), CuO (Copper(II) oxide), FeO (Iron(II) oxide), MnO (Manganese(II) oxide), NiO (Nickel(II) oxide), and ZnO (Zinc oxide) [158]. AgNPs can stimulate wheat growth and yield. Soil applied with 25 ppm of SNPs demonstrated highly favorable growth-promoting effects on wheat growth and yield.

### 5.5. AgNPs for Controlling Viruses in Plants

Naturally occurring nanomaterials are plant viruses, particularly spherical viruses. *Satellite tobacco necrosis* viruses measuring just 18 nm in diameter are the smallest plant viruses known to date [159]. Plant viruses are formed as a genome of a single or double-stranded RNA/DNA encapsulated by a protein coat. Their ability to infect, deliver the genome of nucleic acid to a particular site in the host cell, replicate, package nucleic acid, and precisely exit the host cell in an orderly way required their use in nanotechnology. Due to slower reaction kinetics, biological synthesis provides particles with a strong control over size distribution and form. Kalimuthu et al. reported that maximum silver-nanoparticle synthesis occurs in the stationary phase of *Bacillus licheniformis* [160]. Additionally, Pugazhenthiran et al. reported that the silver-resistant *Bacillus* sp., when cultured with silver nitrate solution, synthesizes AgNPs in the periplasmic space of the cell [161]. Further, an extract of algae was involved in the shape-controlled synthesis of Ag nanoplates due to the different proteins present in the extract [162]. In one additional study, AgNPs exhibited strong inhibitory activity against *Alternaria alternata*, *Botrytis cinerea*, *Curvularialunata*, *Macrophomina phaseolina*, *Rhizoctonia solani*, and *Sclerotinia sclerotiorum* between a 5–15 mg plate^−1^ of potato-dextrose agar [163]. When compared to commercially available AgNPs, the homemade AgNPs at 3 ppm were more active against *Alternaria* sp., *Aspergillus niger*, *Botrytis cinerea*, *Penicillium expansum*, and *Rhizopus* sp. [164].

**Figure 3 jfb-14-00047-f003:**
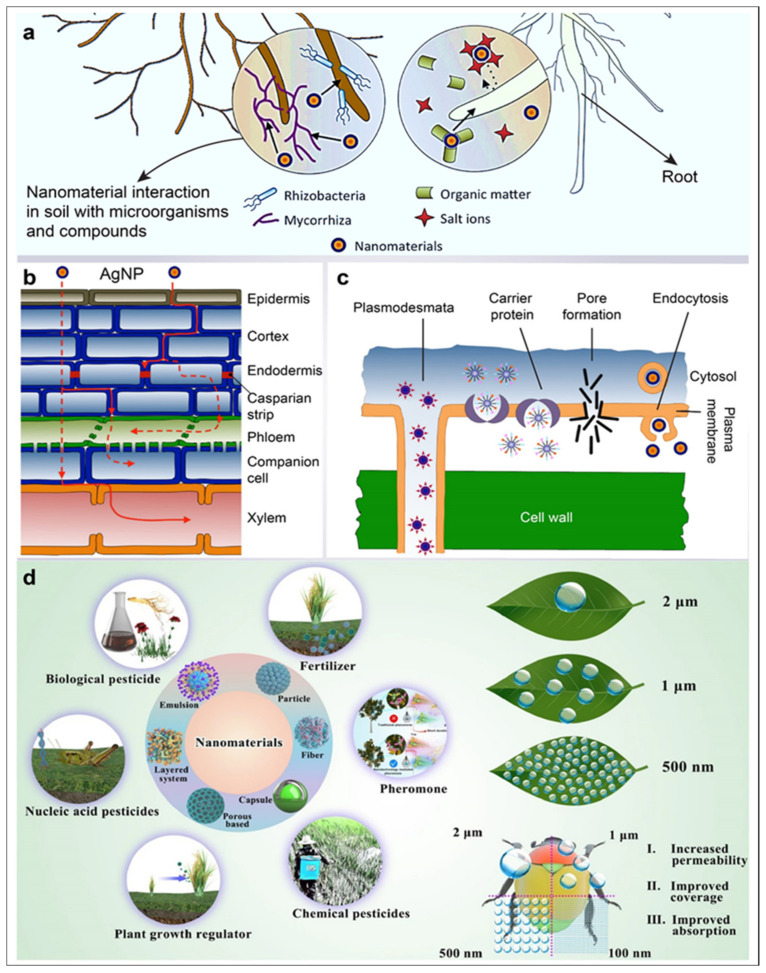
Silver nanoparticle uptake in plants and their application in agriculture. (**a**) AgNP nanomaterial absorption hampers soil due to its interaction with several microorganisms and compounds; (**b**) apoplastic (red solid line) and symplastic (red dotted line) pathways are being followed by nanomaterials for moving up and down; (**c**) penetration of the plant cell (images **a**, **b**, and **c** are adapted from [165]); and (**d**) applications of nanomaterials, including AgNP, for various agriculture purposes. The small size of nanoparticles improves the bioavailability of active ingredients (Image adapted from [166]).

## 6. Influence of AgNPs on the Food Industry

Silver nanoparticle materials have been experiemented with in food applications to figure out the outcome of AgNPs’ antimicrobial properties on food shelf life. “Edible antibacterial films” are formed when food is curved into alginate solutions containing biosynthesized AgNPs in sterilized pears and carrots. Treated pears and carrots found greater customer loyalty over a period of 10 days and provided a lower volume of water [167]. The shelf life of a fresh asparagus spear was extended to 25 days when layered with AgNPs and demonstrated a greener color, little weight loss, and a moderate development of microorganisms during this period [168]. AgNP bags for food storage demonstrated less browning; thicker, firmer, and slower ripening; and a slower decomposition rate [168]. Orange juice stored at 4 °C in a TiO_2_-nanosilver mixture bag showed a substantial decrease in *Lactobacillus plantarum* growth [169]. In another use of AgNPs, Fernández et al. found that the cellulose pads containing AgNPs produced from silver ions in situ have been shown to reduce the microbial levels of atmospheric exudate packaging [170]. The fresh-cut melon had longer lead times for microbial growth and had lower microbial counts on cellulose pads containing AgNPs. Stored fruits in the presence of AgNPs have longer ripening periods. Cellulose pads containing AgNPs formed from silver ions in situ have been shown to reduce the microbial levels of beef-meat exudate processed in changed atmosphere packaging in another form of use. Correspondingly, the fresh-cut melon stored on AgNP-containing cellulose pads had lower microbial counts (mesophiles, psychrophiles, and yeasts) and an increased microbial-growth lag time [170]. Furthermore, the investigation highlighted that AgNPs catalyze ethylene gas degradation and fruits deposited in the presence of AgNPs have slower maturation times and thus an extended shelf life [170] (Table 2).

## 7. Toxicity-Based Investigations of AgNPs

Toxicity profiling of complex substances is the parameter that assists in the determination of the biocompatibility factor of a material. Silver has been obstructed as an antimicrobial agent by its possible toxic effects, and the inappropriate deposition of silver is attributable to issues such as argyria and permanent skin and eye pigmentation. Thus, for consumer goods and before widespread health uses, the toxicity of AgNPs must be researched. To prevent ecological disruption, environmental emissions should also be considered. AgNPs most commonly penetrate the human body via the lung, gastrointestinal tract, scalp, female genitals, and systemic management [188]. Kim et al. performed an analysis of oral toxicity in Sprague Dawley (SD) rats: the results exhibited the accumulation of AgNPs in the blood, liver, lungs, kidneys, stomach, testes, and cortex, but no major genotoxicity activity was observed following the oral administration of AgNPs at different doses of 60 nm of average size for 28 days [128]. Similarly, another study on inhalation toxicity of 1, 98–64, and 9 nm AgNPs in vivo in SD rats did not observe any effect on respiratory, hematological, or blood–biochemical values [189]. In the in vivo cytotoxicity investigations performed on SD rats, major cytotoxicity was observed in the bone of pre-osteoblast MC3T3-C1 cells, as well as toxicities of 3D, porous, 43 μm thick PLGA coatings consisting of up to 2% AgNPs and PLGA (Poly (dl-lactic-co-glycolic acid)) materials during bone regeneration over eight weeks [190]. Other research has shown that, when delivered at a dosage of 20–25 nm, AgNPs successfully inhibit bacteria but induce no cytotoxicity. In animals, these nanoparticles are nontoxic when administered orally, ocularly, or dermally [191]. The study showed that long-term exposure to 90 nm AgNPs induced an inflammatory response in the alveoli and caused lung function changes at all exposure stages [192]. AgNPs have also been shown to move through the blood–brain barrier in rats and cause neuronal degeneration and necrosis over a long time due to accumulation in the brain [193].

Furthermore, in addition to these findings, Vandebriel et al. observed that a functioning immune system was suppressed by the use of both 20 and 100 nm AgNPs in a 28-day, repeated-dose toxicity study in SD rats; in this study, however, AgNPs were provided intravenously [194]. A study published in the field of development reported that very small AgNPs (less than 12 nm in size) in water had adverse effects on the early development of fish embryos, including chromosomal defects and DNA damage, and induced proliferation arrest in zebrafish cell lines, which suggested that human exposure to very small AgNPs could be capable of causing such teratogenic effects [195]. An in vitro study showed that AgNPs are cytotoxic to different types of cells, including cells from the human endocrine system, germ cells, and skin, liver, kidney, lung, and brain [196] cells. In a recent investigation by Panda and Kumari et al., a novel combinatorial technique was used to investigate the function of various AgNP–protein interactions via the first-principle density functional theory and an in silico analysis [197]. From such investigations, it is possible to speculate that nanoparticles created using plant proteins in a newer method will be more biocompatible and environmentally friendly. Since enormous heterogeneity occurs in the particle size, aggregation, and concentration or coating thickness of AgNPs, there is controversy about their total toxicity to humans [198]. Even longer and more comprehensive studies must be carried out to better resolve the risk of AgNP toxicity in humans. Reactive oxygen species are known to increase oxidative stress, which is likely to contribute to the cytotoxicity of AgNPs [199]. AgNPs also weakened cell components, which are further harmed by DNA damage and the activation of antioxidant enzymes. Hepatitis (HepG2 and Caco2) colon carcinoma, which was deficient in antioxidant molecules, showed no AgNP-induced oxidative stress if exposed to AgNP at 1–20 μg/mL [200]. Such observations indicate that the reported cytotoxicity of AgNPs in the HepG2 and Caco2 cells has no important effect on cell-oxidative stress and that a different mechanism of mitochondrial damage mediated by AgNPs contributes to cytotoxicity. The targets for AgNPs appear to be HepG2 and Caco2 cells, which suggests that differences between toxicity mechanisms induced by AgNP are largely due to cell type [200].

In a recent study, the effects of intra-articular injection of AgNPs on rat reproduction and health were assessed, as well as the biodistribution of the AgNP in the development of a contraceptive nanotechnological agent for male animals. In each testicle, treated animals were treated with 220 μL (0.46 μg mL^−1^) of AgNP solution and euthanized 7, 14, 28, and 56 days post-injection. Compared to the control, there was a considerable decrease in the percentage of mobile sperm in D7 (8.8 percent). All animals had hematologic parameters, creatinine, and alanine aminotransferase levels within normal limits for Wistar rats. There was always a higher percentage of silver in the liver than in the others. Some acute and serious toxic effects have been observed in sperm cells following ta reversible, intra-specular injection of AgNPs. To further assess the possible antioxidant effect of ascorbic acid, a different group of scientists in another recent study looked at the toxic effect of AgNPs on the parotid glands (PG) of albino rats histologically and ultrastructurally. They found that there was a significant pathological change as well as a potential antioxidant effect. There were three groups of thirty male, albino rats weighing between 150 mg and 200 mg: ten rats received 2 mg cg^−1^ (body weight of intraperitoneally injected nitrate buffer (IP) daily for 28 days, ten rats received 2 mg kg^−1^ of IP AgNP daily for 28 days, and ten albino rats received 2 mg kg^−1^ of IP AgNP containing vitamin C daily for 28 days. In the group of PG acinar and ductal cells from AgNPs, signs of toxicity and degeneration were shown to be characterized by pleomorphic nuclei, binucleate, cytoplasmic vacuolations, and stagnant secretion. A dilated raw endoplasmic reticular and lysosomes, in addition to degenerated mitochondria, were also completed with AgNP (*p* < 0.001). In comparison with the AgNP–vitamin C Group, the histological and ultrastructural changes were significantly lower (*p* = 0.002). AgNPs produced a considerable toxic effect on PG for albino rats, probably by creating a reactive species of oxygen and releasing toxic ions. Additionally, vitamin C has been demonstrated effectively to reduce these toxic effects [201]. It is speculated that the inhibitory effects of AgNPs on mitochondria are linked to the size of the nanoparticles, with smaller AgNPs (15 nm) being more toxic than larger ones (55 nm) [202]. When these factors are all taken into consideration, the oxidative stress, cell type, and the method of AgNP administration all contribute to toxicity. AgNPs have a significant impact on the microbial community and cause the natural seasonal progression of tundra assemblages to be halted [203]. This effect was evidenced by levels of differential respiration in 0.066% AgNP-treated soil—which was only half the levels of control soils—a decrease in signature bacterial fatty acids, and changes in both richness and evenness in bacterial and fungal DNA-sequence assemblages. Similarly, Volker et al. investigated the effects of AgNPs on *Potamopyrgus antipodarum*, a freshwater mudsnail, and found that AgNPs decreased their reproduction, thereafter reducing their population growth [204]. Thus, further assessment of the environmental pollution and risks due to AgNP exposure is required to prevent damage to the environment (Figure 4).

## 8. Prospects

The use of AgNPs is already established for many commercial applications and certain medical applications, such as wound dressings, while many new potential applications are being heavily investigated. AgNPs possess great potential due to their antibacterial, antifungal, antiviral, and anti-inflammatory properties, while our recent research has revealed novel osteo-inductive properties as well. However, the mechanisms and biological interactions behind these properties are not fully understood. For example, the relationship between the size and shape of AgNPs and their biological properties and toxicity is not properly revealed and thus requires further investigation as AgNP use continues to increase. Therefore, there is a pressing need to fully elucidate the mechanisms concerning the efficacy and toxicity of AgNPs before their widespread medical application can occur.

## 9. Conclusions

AgNPs are an extensively explored nanomaterial that is being synthesized chemically as well as by using biological entities that mainly include plants and microbes. Several techniques are available to monitor their synthesis and to characterize their shape and size. Parameters such as size, shape, concentration, surface charge, and colloidal state are crucial for their functionality and determine their physicochemical and biological properties. These biological properties are crucial for various biomedical applications that include significant antimicrobial, anti-cancer, anti-inflammatory, and anti-diabetic activities, etc. Apart from these applications, the use of AgNPs in wound dressing, bone cement, and dental caries enhances the spectrum of their applications. Their unique mechanical, optical, and chemical attributes are being used in designing devices for the diagnosis of diseases such as Alzheimer’s disease. The role of AgNPs in the food, cosmetic, and textile industries is opening avenues for its widespread industrial utilization.

The promising and extraordinary properties of this multi-functional nanomaterial have scope in future medical, biomedical, and other applications of great human significance. However, exhaustive investigations are required to assess their short-term and long-term toxicity to be enable their proper application for human benefit. Moreover, the surface modification and functionalization of these particles may pave the way for the investigation of their novel applications and the improvement of the already established applications. In light of the aforementioned investigations, AgNPs have established themselves as potential and effective agents for a variety of human-benefiting applications.

The positive impacts of AgNPs as novel, biocompatible, and nanostructured materials created for contemporary treatment techniques are supported by a sizable body of study data. AgNPs offer additional mechanical, optical, chemical, and biological peculiarities in several beneficiary applications. In addition, their attractive and versatile antimicrobial potential makes them a good choice for the design, acquisition, assessment, and clinical evaluation of performance-enhanced biomaterials and medical devices. However, in-depth studies on their immediate and long-term toxicity, as well as the toxicity-related mechanisms at play, are needed. Lastly, a deeper comprehension of the mechanisms adopted by AgNPs justifies further study to increase the scope of their nanomedical consumption in agriculture, pharmaceutics, therapeutics, and diagnosis.

## Figures and Tables

**Figure 4 jfb-14-00047-f004:**
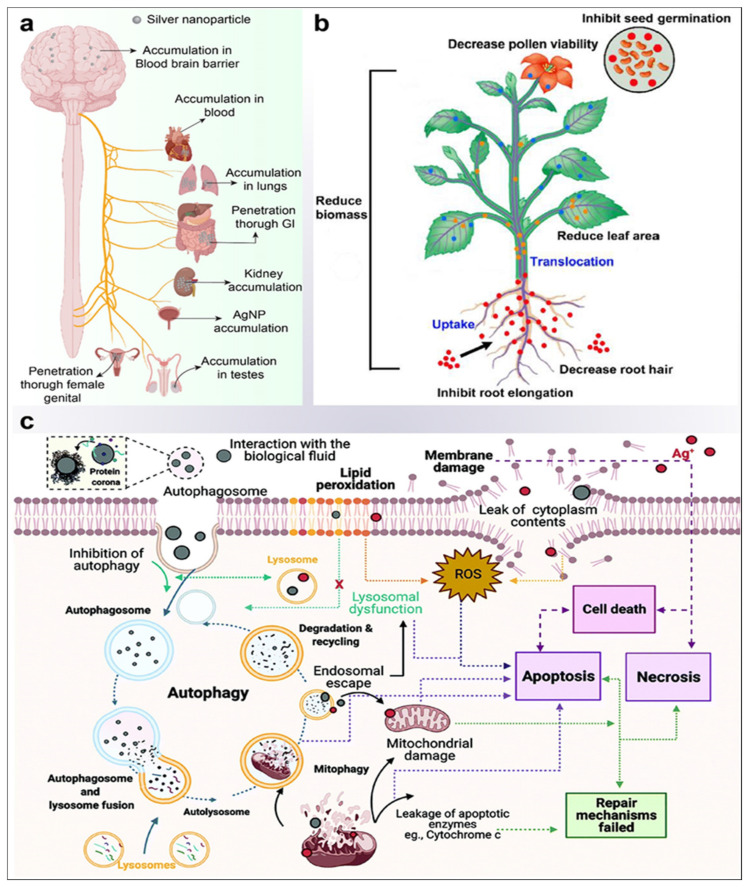
Cytotoxicity of the silver nanoparticle in plants and animals and its mechanism (**a**) Accumulation and penetration of AgNPs in different organs including the blood–brain barrier; (**b**) uptake, translocation, and major phytotoxicity of AgNPs in the plant (image adapted and modified from [205]); and (**c**) possible cytotoxicity molecular mechanism induced by AgNPs. Autophagosome is strongly linked with the protein corona formation for uptake of AgNPs in the cell (Image adapted and modified from [206]).

**Table 2 jfb-14-00047-t002:** Application of AgNPs in the field of agriculture and in the food industry.

S. No.	Source of Synthesis	Size Range of AgNPs (nm)	Agriculture and Food Industry Applications	References
1.	*Shewanella algae bangaramma*	5–30 nm	Pest control	[171]
2.	β-1, 3 glucan	32–46 nm	Nanofertlizer	[172]
3.	*Allium cepa*	10–30 nm	Nanofertilizer	[173]
4.	Rice starch	8 nm	Effect on agronomic traits of onion	[174]
5.	Tri-sodium citrate	20 nm	Effect on growth, some biochemical aspects, and yield of *Trigonella foenumgraecum*	[175]
6.	*Tagetes erecta*	60 nm	Effect on the growth of *Zea mays*	[176]
7.	*Serratia* sp. *BHU-S4*	10–20 nm	Effect against spot blotch disease in wheat	[177]
8.	Jack fruit seeds	20–30 nm	Antibacterial activity against food-borne bacteria	[178]
9.	*Commercially purchased*	10 nm	For increasing the shelf life of fresh orange juice	[179]
10.	Electrostatic adsorption method	10 nm	Ant mildew and in the storage of rice	[180]
11.	Fungal extract	3–20 and 4–20 nm	Impact on Seed Germination and Seedling Growth	[181]
12.	Chemical reduction	30 and 40 nm	Effect on plant growth and soil bacterial diversity	[182]
13.	*Coriandrum sativum* leaves extract	20–80 nm	Effect on the growth potential of *Lupinus termis* L. seedlings	[183]
14.	*Aloe vera* leaves	47 nm	Phytotoxic impact on *Brassica* sp.	[184]
15.	*Berberis lycium*	54 nm	Impact on protein and carbohydrate contents in seeds of *Pisum sativum*	[185]
16.	*Commercially purchased*	50–80 nm	Soil improvement	[186]
17.	*Raphanus sativus* var. *aegyptiacus*	2–100 nm	Control of land snail *Eobania vermiculata* and pathogenic fungi on plants	[187]

## Data Availability

All data included in this study are available upon request from the corresponding author.

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
