# Peer review of "Emerging Trends in Advanced Translational Applications of Silver Nanoparticles: A Progressing Dawn of Nanotechnology"

_jfb, 2023, doi:10.3390/jfb14010047_

Round 1

Reviewer 1 Report

Manuscript title:  Emerging trends in advanced translational applications of Silver Nanoparticles: A progressing dawn of nanotechnology

 Journal:      JFB (ISSN 2079-4983)

Manuscript ID: jfb-2134396

Dear Editor: 

Thank you to send me this manuscript. I wrote my evlution below:

(1) Present original findings, conclusions or analysis that has not been published previously by the authors or others : Yes

  (2) Written clearly: No

 (3) Have a high impact in its subfield: Yes

This manuscripte (Emerging trends in advanced translational applications of Silver Nanoparticles: A progressing dawn of nanotechnology). The work presented here is interesting and scalable. It is a very relevant subject of study. I would recommend publication of this manuscript in this journal after the authors rigorously addressed some comments listed below.

1)      The abstract section does not have all important information. I prefer to rewrite this section by fiiling the spesefice and importante informations. So, it has more confuse for me.

2)      The introduction section has a very good information, the authors do the best for this section.

3)        There are several passages of the manuscript which are incorrect.

4)      The references included in terms of the system studied are outdated, I encourage the authors to make a new search to include more recent references from the last 5 years .

Finally, if the authors do all these comments, I recommend to publish this manuscript in this journal.

        Best Regards

Assist. Pro. Dr.Mohammed H. Mohammed

Author Response

Reviewer 1:

This manuscripte (Emerging trends in advanced translational applications of Silver Nanoparticles: A progressing dawn of nanotechnology). The work presented here is interesting and scalable. It is a very relevant subject of study. I would recommend publication of this manuscript in this journal after the authors rigorously addressed some comments listed below.

1) The abstract section does not have all important information. I prefer to rewrite this section by fiiling the spesefice and importante informations. So, it has more confuse for me.

Response: We thank reviewer for raising the comment. We have rewritten the abstract.

2)The introduction section has a very good information; the authors do the best for this section.

Response: We thank reviewer for the appreciation.

3) There are several passages of the manuscript which are incorrect.

Response: We thank reviewer for deep review of the manuscript. We have revised the whole manuscript and tried to make it error free.

4)The references included in terms of the system studied are outdated, I encourage the authors to make a new search to include more recent references from the last 5 years.

Response: We thank reviewer for raising the comment. We have rearranged the references as per the reviewer’s suggestion.  

Reviewer 2 Report

The manuscript by Husain et al. delivers a comprehensive and timely review of the advanced translational applications of silver nanoparticles. Important or foundational as the fundamental sciences of nanotechnologies can be, the real-world impact is still the main driving force for developing nanotechnologies’ future. As nanotechnology emerges, evolves, and becomes inseparable from our daily lives, it is high time to take a step back and see the bigger picture. The authors gathered and refined their examples into easily understandable pieces, encouraging researchers both within and outside the field of Nano to think about what opportunities lie ahead. As I find this review could serve as excellent reading material, I recommend acceptance with minor grammatical edits, such as the following –

  1. The overall English of this manuscript can be improved, mainly to avoid sporadic casual tones such as Line 18-19.
  2. In line 13, surface surface area
  3. In line 21, the most one of the most, other types of nanoparticles such as silica or gold are comparatively investigated as nano silver, perhaps not as widely applied

Author Response

The manuscript by Husain et al. delivers a comprehensive and timely review of the advanced translational applications of silver nanoparticles. Important or foundational as the fundamental sciences of nanotechnologies can be, the real-world impact is still the main driving force for developing nanotechnologies’ future. As nanotechnology emerges, evolves, and becomes inseparable from our daily lives, it is high time to take a step back and see the bigger picture. The authors gathered and refined their examples into easily understandable pieces, encouraging researchers both within and outside the field of Nano to think about what opportunities lie ahead. As I find this review could serve as excellent reading material, I recommend acceptance with minor grammatical edits, such as the following –

1.The overall English of this manuscript can be improved, mainly to avoid sporadic casual tones such as Line 18-19.

Response: We thank reviewer for raising the comment. We have revised the whole manuscript and tried to make it error free.

2.In line 13, surface  surface area

Response: We thank reviewer for raising the comment. The text has been edited.

3.In line 21, the most  one of the most, other types of nanoparticles such as silica or gold are comparatively investigated as nano silver, perhaps not as widely applied

Response: We thank reviewer for raising the comment. The text has been edited.

Reviewer 3 Report

The manuscript provides insights into various biomedical applications of silver nanoparticles.

I suggest the following minor revision before considering this manuscript again for publication.

Please find below my comments:

1.     In the abstract section, line 16-17 discuss about the green synthesis of silver nanoparticles, provide a proper reason for its popularity and reframe the sentence.

2.     Nanomaterials are easier to be consumed, resulting in increased use of human cells, tissues, and organs (line 17-18). The above statement intent is not clear, reframe the sentence.

3.     Line 51-56 provides information about biological approaches of synthesizing silver nanoparticles. But it is represented as chemical approaches, clarify the same.

4.     In section 2, provide in-depth discussion about the advantages and limitations of different techniques of nanoparticle synthesis.

5.     Provide in-depth discussion about the mechanism of silver nanoparticles with relevance to various biomedical applications along with diagrammatic representation.

6.     In the section of AgNPs as an anti-inflammatory substance, the information provided is not clear. Reframe the entire section with proper cross-citations.

7.     AgNPs as an odor-controlling agent in the textile industry section is not clear. Reframe the entire section with proper cross-citations.

8. The manuscript should be checked thoroughly for grammatical mistakes.

Author Response

The manuscript provides insights into various biomedical applications of silver nanoparticles. I suggest the following minor revision before considering this manuscript again for publication.

Please find below my comments:

  1. In the abstract section, line 16-17 discuss about the green synthesis of silver nanoparticles, provide a proper reason for its popularity and reframe the sentence.

Response: We thank reviewer for raising the comment. We have rewritten the abstract. Line 16-17 has been edited.

  1. Nanomaterials are easier to be consumed, resulting in increased use of human cells, tissues, and organs (line 17-18). The above statement intent is not clear, reframe the sentence.

Response: We thank reviewer for raising the comment. We have rewritten the abstract. Line 17-18 has been reframed.

  1. Line 51-56 provides information about biological approaches of synthesizing silver nanoparticles. But it is represented as chemical approaches, clarify the same.

Response: We ask apologies for the inconvenience caused due to mistakes. We have revise the section and clarified the biological synthesis in the text.

  1. In section 2, provide in-depth discussion about the advantages and limitations of different techniques of nanoparticle synthesis.

Response: We thank reviewer for the suggestion. We have revised the section as per the suggestion.

  1. Provide in-depth discussion about the mechanism of silver nanoparticles with relevance to various biomedical applications along with diagrammatic representation.

Response: The mechanism of the silver nanoparticles for biomedical application has been included in the text and has been diagrammatically represented in figure 2 and figure 4.

  1. In the section of AgNPs as an anti-inflammatory substance, the information provided is not clear. Reframe the entire section with proper cross-citations.

Response: We ask apologies for the inconvenience caused due to mistakes. We have reframed the section.

  1. AgNPs as an odor-controlling agent in the textile industry section is not clear. Reframe the entire section with proper cross-citations.

Response: We have reframed the entire section.

  1. The manuscript should be checked thoroughly for grammatical mistakes.

Response: We thank reviewer for the suggestion. We have revised the manuscript and hope that the reviewer will find it as per the expectation.

Reviewer 4 Report

In this manuscript, the authors summarized modern applications of silver nanoparticles including their medical use, anti-bacterial and anti-viral applications, etc. The synthetic routes and characterization were also substantially introduced. Overall, this review is well-organized and provided valuable information of recent progress of silver nanoparticle studies. There are some minor issues to be addressed before acceptance of publication.

1.     Some statements in this manuscript are a little bit arbitrary or absolute. For example, in the abstract the authors mentioned “Nano-silver is the most investigated and used nanoparticle.” The authors should revise with more rigorous description.

2.     Several types of AgNPs could be used as contrast agents for photoacoustic imaging and computed tomography imaging for disease diagnosis or visualization. The authors might consider mentioning these applications in the manuscript.

3.     AgNPs also play important roles as photoactive agents, drug delivery carrier, or radiosensitizer in cancer treatments. These applications should be included in section 4.1.10 Anti-cancer progressions of AgNPs.

Author Response

Reviewer 4

In this manuscript, the authors summarized modern applications of silver nanoparticles including their medical use, anti-bacterial and anti-viral applications, etc. The synthetic routes and characterization were also substantially introduced. Overall, this review is well-organized and provided valuable information of recent progress of silver nanoparticle studies. There are some minor issues to be addressed before acceptance of publication.

  1. Some statements in this manuscript are a little bit arbitrary or absolute. For example, in the abstract the authors mentioned “Nano-silver is the most investigated and used nanoparticle.” The authors should revise with more rigorous description.

Response: We thank reviewer for valuable comments and suggestion. As suggested, we have revised the whole section.

  1. Several types of AgNPs could be used as contrast agents for photoacoustic imaging and computed tomography imaging for disease diagnosis or visualization. The authors might consider mentioning these applications in the manuscript.

Response: We thank reviewer for the suggestion. We have revised the section as per the suggestion.

  1. AgNPs also play important roles as photoactive agents, drug delivery carrier, or radiosensitizer in cancer treatments. These applications should be included in section 4.1.10 Anti-cancer progressions of AgNPs.

Response: We thank reviewer for the suggestion. We have revised the section as per the suggestion.

Reviewer 5 Report

The review is well-written and interesting, but you should implement the originality of this work.

Compared to other recent studies on AgNP applications, what more does this review describe?

Author Response

The review is well-written and interesting, but you should implement the originality of this work. Compared to other recent studies on AgNP applications, what more does this review describe?

Response: We heartily thank reviewer for the deep review of our manuscript and the valuable suggestion offered. We have revised the manuscript including more details in each section. The review was intended to present the advances in synthesis and modern applications of AgNPs. It also lists in detail about the biomedical and ecological applications as a for enlightening the future applications.